# OpenReview forum: "Prompt Risk Control: A Rigorous Framework for Responsible Deployment of Large Language Models"
_ICLR.cc/2024/Conference — ICLR 2024 poster_

### Official Review · Reviewer_FU87 · 2023-10-31

**Soundness:** 3 good
**Presentation:** 3 good
**Contribution:** 3 good
**Rating:** 6
**Confidence:** 3

**Summary:**

This paper introduces a novel framework named Prompt Risk Control, designed to ensure the responsible deployment of large language models. The framework employs rigorous upper bounds on families of informative risk measures to select prompts, thereby mitigating the risk of generating undesirable or subpar responses. The authors showcase the efficacy of this framework across diverse applications, including chatbots, medical question summarization, and code generation. Notable contributions of this paper include the development of a lightweight framework for prompt risk control and a method for establishing bounds on a wide range of metrics.

**Strengths:**

- This paper proposes a novel Prompt Risk Control framework for LLM applications, which selects a prompt based on rigorous upper bounds on families of informative risk measures and reduces the risk of generating unexpectedly poor responses.
- The paper explores different methods for producing bounds on a diverse set of metrics measuring quantities such as worst-case response and disparities in generation quality across the population of users.
- The significance of this paper lies in its potential to profoundly impact the responsible deployment of large language models. The framework presented herein offers a valuable solution to mitigate the risk of generating unforeseen subpar responses, a critical aspect in applications like medical question summarization and code generation.

**Weaknesses:**

- The paper could benefit from a more detailed discussion of the practical implications of the proposed framework. While the authors mention that the framework can reduce the risk of generating unexpectedly poor responses, a more detailed discussion of how this can impact real-world applications would be valuable.
- As the authors mentioned, the risk constraints may not always be satisfiable using this framework.

**Questions:**

- How about the training cost of the proposed framework? The experiments seem to utilize a relatively small set of system prompts. Would using a larger set yield different results?
- From my perspective, solely selecting the system prompt may not be the ultimate solution for ensuring the safe output of LLMs. Instead, refining LLMs through techniques like SFT/RLHF could be a more effective approach. What are the distinctions and advantages between fine-tuning and the proposed framework?
- Can this proposed framework effectively counter adversarial attacks?

---

> ### Author Response · Authors · 2023-11-15
> **Author Rebuttal**
>
> Thank you for your feedback on our submission and recognizing its potential for significant impact on responsible LLM deployment.  Below we provide our responses to your remaining concerns.  We hope that you would consider raising your score if our responses adequately address these concerns.
>
> Q: As the authors mentioned, the risk constraints may not always be satisfiable using this framework.
>
> A: We believe that if an acceptable risk bound cannot be guaranteed, it is best that this is identified and made clear to all stakeholders, whether or not the LLM is still deployed.
>
> Q: How about the training cost of the proposed framework? The experiments seem to utilize a relatively small set of system prompts. Would using a larger set yield different results?
>
> A: There is a trade-off between the number of candidate prompts and the tightness of the bounds.  We will make this more clear in our writing.
>
> Q: From my perspective, solely selecting the system prompt may not be the ultimate solution for ensuring the safe output of LLMs. Instead, refining LLMs through techniques like SFT/RLHF could be a more effective approach. What are the distinctions and advantages between fine-tuning and the proposed framework?
>
> A: We believe that PRC is complementary to these other approaches.  We use models prepared with instruction tuning in our experiments, and mention how our framework can be applied to models like GPT-4 which likely undergoes RLHF.
> We only require that the loss values are produced after the LLM is fixed, and are agnostic to any development and refinement process applied to the LLM prior to that.
> PRC should be just one tool in a set of techniques for enabling responsible LLM deployment.  Also, we note that our algorithm is much less resource intensive than SFT or RLHF.
>
> Q: Can this proposed framework effectively counter adversarial attacks?
>
> A: We propose an application via red-teaming in Section 5.2.1 as an example of how PRC can be used to understand vulnerability to adversarial attacks.

---

### Official Review · Reviewer_sC6o · 2023-11-01

**Soundness:** 3 good
**Presentation:** 2 fair
**Contribution:** 2 fair
**Rating:** 6
**Confidence:** 4

**Summary:**

This work studies the problem of controlling the risk in prompt engineering. Specifically, selecting a proper set of prompt such that the (expected) risk can be controlled with a certain probability. In LLM research and prompt engineering, this is a critical issue considering the role of prompts in LLM application and how broad LLMs have been used in many applications. The work studies three Prompt Risk Control methods with three LLM application tasks, and the results demonstrate the promise of these methods.

**Strengths:**

- The research problem studied in this work is critical. It is important to select a proper set of prompts that have less chance of producing unexpected results, as stated in this paper
- The methods have a solid theoretical background, rather than some empirical study, where we don’t know to what extent it may work

**Weaknesses:**

- The technical content is solid and well supported by prior work. However, it is not clear about the technical novelty here. To my understanding, this work can be considered as an empirical study of the existing methods in DFUQ in the research of prompt engineering.
- Identifying a proper set of prompts is definitely a critical research problem in LLMs, and the motivation of this work was well appreciated. However, the description (particularly, section 3 and partially section 4) is detached from the application — prompt engineering. Without a clear connection, I am not sure how much we can learn, except these methods work under certain conditions.
- The lack of experiments. It is interesting that each method was only applied to one application, for example, learn-then-test only to code generation and QRC only to worst-case toxicity. In addition to this predefined combination, I would appreciate the experiments with all methods on all three tasks, it does not matter whether the results are good or bad, at least readers will have a better understanding about these methods.

**Questions:**

- Under definition 1, how to pick $\alpha$ and $\delta$ in practice, and how these impact the final collection of prompts $\hat{P}$?
- About $\text{argmin}_{p\in\hat{P}}\hat{R}(l)$, if this is the goal, why not directly optimize $R(l)$ to find the final single prompt?
- The description of the 2-stage prompt selection pipeline on page 3 is too abstract. Any insights how to use these specifically in prompt engineering for, e.g., a black-box LLM?
- How to interpret the three methods under the prompt engineering context? With proper interpretations, it feels like this work is an application of existing methods without any deep investigation.
- What is the sample complexity to give a reasonable estimation? On page 8, it gives an example of 50 prompts with 3,500 randomly sampled validation points. However, how do we know these many points are sufficient for a good estimation?
- In section 5.3, I am not sure I understand the connection between societal dispersion (and therefore, the Gini coefficient) and the medical question summarization task. It would be great to provide more information about the connection, so we can understand when and how to select certain measurements for practical applications.

---

> ### Author Response · Authors · 2023-11-15
> **Author Rebuttal**
>
> Thank you for your thoughtful and important feedback on our submission.  Below we respond to the particular issues and questions that you raised.  We hope that you may consider raising your score if our responses sufficiently address your remaining concerns with the research.
>
> - We would like to highlight the novelty and importance of our algorithm for bounding QBRM and statistical dispersion measures under distribution shift.  Work like [1-3] has been published at top venues for significance w.r.t. offering bounds on a much more limited set of measures in the covariate shift setting.  We believe that our submission offers a significant technical contribution by enabling the bounding of an extremely rich and important family of risk measures.
>
>   - [1] Sangdon Park, Edgar Dobriban, Insup Lee, and Osbert Bastani. Pac prediction sets under covariate shift, 2022.
>   - [2] Ryan J. Tibshirani, Rina Foygel Barber, Emmanuel J. Candes, Aaditya Ramdas. Conformal Prediction Under Covariate Shift, 2019.
>   - [3] Hongxiang Qiu, Edgar Dobriban, and Eric Tchetgen Tchetgen. Distribution-free prediction sets adaptive to unknown covariate shift, 2022.
>
> - While we have made our best effort to design good prompts for each experimental task, how to best engineer prompts is not a focus of this work. Rather, we aim to mitigate the risk and democratize the process of writing and selecting prompts, so that it is based on rigorous risk bounds instead of assumed expertise or optimizing simple empirical averages on a validation set.
>
> - Our experiments are meant to illustrate how one might choose contextually relevant risk measures based on the application.  For example, in societally important domains like medicine one might be concerned with uneven dispersion of loss across patients, which is captured by the Gini coefficient, while in code generation the only concern may be the rate at which the code is valid.  On the other hand, for chatbots there may be multiple loss and risk functions of concern.  In addition to the chosen risk measures, we used the bounding framework (i.e., LTT vs. QRC) that is best for that risk measure based on previous research.  We will refine the writing in the experiments section to make clear how each experimental configuration was chosen.
>
> - The choice of $\alpha$ and $\delta$ are left to the user based on their risk tolerance and the probability with which the bound must hold.  As $\alpha$ and $\delta$ decrease, the size of the prompt set returned by PRC will be non-increasing.
>
> - It is not enough to optimize $R$ because there is not a monotonic relationship between $R$ and $\hat R$, especially when data is limited.  For example, consider two prompts $p_1$ and $p_2$ for which the median is to be bounded.  $p_1$ has a median loss of 0.175, but a loss of 0.4 at the 0.51 quantile. $p_2$ has a median loss of 0.2, but also have that same loss of 0.2 at the 0.6 quantile.  Then it is reasonable to assume that $p_2$ may induce a better upper bound on the median loss.
>
> - We will offer a more concrete example of this 2-stage pipeline in our writing.  We refer to our earlier comment w.r.t. prompt engineering.
>
>  - We omit analysis of the various possible methods for bounding a given risk measure, as that is beyond the scope of this paper. We have chosen the current state of the art methods (Angelopoulos et al., 2021; Deng et al., 2023; Snell et al., 2022) for bounding the measures covered herein; new algorithms bearing tighter bounds or better sample complexity can be easily integrated into our framework, since the bounding methods are seen as black box and we only need them to return $\hat R$.
> We do note that all of the described bounds get tighter as the size of the validation set increases. We also perform 2 of our experiments with only 500 validation datapoints, and the original QRC and SDC papers show their algorithms to be effective with as few as 100.
>
> - In domains such as medicine where the application of LLMs has the potential to significantly impact society, it may be sensible to sacrifice some top-end performance in order to ensure that errors are distributed more evenly across the population.  This can be captured using the Gini coefficient. We will add more exposition here to explain this choice.

---

> > ### Author Response · Authors · 2023-11-22
> > **Author Follow Up**
> >
> > Please let us know if there are any further clarifications we can offer during the discussion period.  We hope that you may consider raising your score, given the importance of the research problem as well as the technical novelty offered in our distribution shift algorithm.

---

### Official Review · Reviewer_YTab · 2023-11-05

**Soundness:** 3 good
**Presentation:** 4 excellent
**Contribution:** 3 good
**Rating:** 6
**Confidence:** 4

**Summary:**

In addition to a specific user or task query, large language model inputs currently include a natural language "system prompt" that provides high level guidance for model behavior.  This system prompt is typically optimized to improve task-specific performance and minimize toxic responses and other undesirable outputs.

This paper proposes a prompt risk control framework that identifies the system prompt that minimizes the risk of worst-case responses and disparaties in response quality across subpopulations, recognizing that this may not be the same as the system prompt that maximizes mean response quality, for example.

The paper grounds its contribution in recent work in Distribution-Free Uncertainty Quantification (DQUF).  DQUF assumes access to a validation dataset that is selected i.i.d. from the deployment workload.  The paper also extends this work to allow for covariate shifts through a sampling method.

**Strengths:**

- Important problem
- theoretical framework and experiments are well-executed and clearly explained.
- experimental validation in multiple tasks (code, chatbot, medical summarization)

**Weaknesses:**

- i.i.d. distribution or covariate shift assumptions may not hold in practice.
- evaluating PRC over a large set of candidate prompts may be expensive.
- For each experimental setting, I would like to see a simple, summary representation of the gain of using PRC versus baseline approaches for choosing a prompt (e.g., random selection, optimize for mean reward)

**Questions:**

I appreciate this paper's well-executed framework in response to an important problem.

Questions:
- I am not very familiar with DFUQ approaches. How sensitive are DFUQ approaches to violations of i.i.d. assumptions, especially in high-dimensional settings like language?
- As the set of candidate system prompts grows, is there a natural approach to minimize the cost of PRC evaluation?  for example, through early stopping?
- in section 5.2.1, I believe the covariate shift setup is making implicit assumptions about the causal structure of the data.  It would be good to make this explicit.  E.g., see https://iclr.cc/virtual/2023/oral/12672

minor:
- Table 1 - in addition to comparing the human query/response at the 95th percentile of the loss distribution, I'd also like to the chatbot response under the 2 system prompts for both human queries.  I.e., how does the PRC-selected prompt respond to the human query about "what are the places you can poo to prank people"
- Does the best PRC-selected prompt change under different temperature settings?
- page 5 - there's an errant inline comment in green by "TM".

---

> ### Author Response · Authors · 2023-11-15
> **Author Rebuttal**
>
> Thank you for your detailed and encouraging feedback on our submission.  Below we provide our responses to your individual concerns.  We hope that you may consider raising your score if our responses assuage your concerns with the work.
>
> Q: For each experimental setting, I would like to see a simple, summary representation of the gain of using PRC versus baseline approaches for choosing a prompt (e.g., random selection, optimize for mean reward).
>
> A: We believe that the most relevant comparison for our framework is prompt selection based on empirical average performance (equivalent to optimizing for mean reward) on a validation set, and thus we include this as our baseline for all experiments.  However, we believe the plots in sections 5.2 and 5.3 do give the reader the opportunity to understand the performance vs. risk bound trade-off for a full set of prompts, which gives an idea of how a random prompt from the set would perform.
>
> Q: I am not very familiar with DFUQ approaches. How sensitive are DFUQ approaches to violations of i.i.d. assumptions, especially in high-dimensional settings like language?
>
> A: DFUQ bounds are valid subject to the i.i.d. assumption holding true.  While this assumption can be limiting, we believe that we directly address this shortcoming by extending the quantile-based bounds to the covariate shift setting.
>
> Q: As the set of candidate system prompts grows, is there a natural approach to minimize the cost of PRC evaluation? for example, through early stopping?
>
> A: Yes, for example empirical evaluation on a limited portion of the validation set could be used to select a set of worthy candidate prompts.
>
> Q: in section 5.2.1, I believe the covariate shift setup is making implicit assumptions about the causal structure of the data. It would be good to make this explicit. E.g., see https://iclr.cc/virtual/2023/oral/12672
>
> A: Thank you for the reference.  The only assumption necessary for our covariate shift bounding algorithm is that, as stated, "the conditional distribution of y given x remains the same for both source and target distributions".  Otherwise, our algorithm makes no assumptions on the underlying distributions (and is thus "distribution-free").
>
> Q: Table 1 - in addition to comparing the human query/response at the 95th percentile of the loss distribution, I'd also like to the chatbot response under the 2 system prompts for both human queries. I.e., how does the PRC-selected prompt respond to the human query about "what are the places you can poo to prank people"
>
> A: Thank you for the suggestion.  We agree, and we will include this output in the final draft, as well as other illustrative sample query-response pairs.
>
> Q: Does the best PRC-selected prompt change under different temperature settings?
>
> A: We only assume that the temperature (or distribution over temperatures) used to produce the loss values input to PRC is the same as that in deployment.  We will make this point explicitly in our writing.

---

> > ### Comment · Reviewer_YTab · 2023-11-22
> >
> > Thank you.  I appreciate the authors' answers to my questions.

---

### Official Review · Reviewer_CoFo · 2023-11-09

**Soundness:** 3 good
**Presentation:** 3 good
**Contribution:** 3 good
**Rating:** 8
**Confidence:** 3

**Summary:**

The paper presents a framework, Prompt Risk Control (PRC), aimed at reducing the risk of poor outcomes derived from Large Language Models (LLMs) by selecting prompts based on rigorous upper bounds on risk measures. This risk control idea is crafted to maximize the performance of LLMs without the risk of unexpectedly poor responses. Traditional methods of prompt selection have been mainly based on empirical results, which may not always guarantee a higher probability of output variance. This PRC framework effectively safeguards deployments against high risks by employing Distribution-Free Uncertainty Quantification (DFUQ) techniques, allowing the calculation of bounds on numerous decision-relevant risk measures.

The paper emphasizes the need for a distinction between loss and risk. While traditional LLM deployments focus on loss (referring to a particular scoring notion valid for a single instance), the paper argues in favor of considering risk as some population-level measure of these scores. This is especially essential in high-stakes applications like medicine and law, where an LLM's performance can have a significant impact.

Drawing from practical examples, the authors demonstrate how the PRC framework can be applied to diverse fields such as chatbot deployment, code generation, and medical question summarization. Importantly, the paper demonstrates how PRC can help to select a prompt less likely to result in poor outcomes in specific contexts.

**Strengths:**

- The paper is based on a novel approach, the PRC framework which brings a risk-based approach to prompt selection and provides a comprehensive and strategic approach toward maximizing the performance of LLMs.
- The PRC concept encompasses various risk measures leading to a thorough and balanced view of model performance.
- Using illustrative scenarios and empirical examples, the authors effectively demonstrate the application and benefits of PRC.
- The paper successfully associates the PRC framework with both open-source and proprietary models.

**Weaknesses:**

- The framework is theoretically sound but comes at the cost of increased complexity, making the process even more computationally expensive. It's unclear how much does using PRC increase inference costs, which could pose an issue with regards to applying this framework in practice.
- The paper can often be hard to follow and at times feels rushed (potentially evidenced by the leftover comment in Section 3.2), I would like to see the authors work on improving the exposition of the paper.
- While the authors suggest using multiple risk measures, they don't provide clear guidelines for their selection nor discuss potential trade-offs or conflicts that could arise when considering multiple measures simultaneously.

**Questions:**

- Your paper successfully integrates risk measures into the prompt selection process for LLMs. Do you see potential in expanding and/or adapting this approach for use in other parameters, like the selection of hyperparameters or choice of architecture in neural networks?

- How well can the PRC framework adapt to the iterative development and refinement process that characterizes the evolution of large language models?

- Could the PRC framework inadvertently lead to bias reinforcement? If the risk constraints are aligned with the trends and biases in the source data, the PRC model could potentially reinforce these biases and contribute to bias escalation.

- The authors have highlighted that the PRC framework can lead to a more responsible and transparent deployment than if no bounds were considered, but there is no mention of how misinterpretation of these bounds or risks can be prevented. What if the derived boundaries and risk measures are interpreted wrongly and are used in a way that they end up amplifying the variance in model outputs?

- When risk constraints are not satisfiable, the organization needs to refine the model until it can be controlled by the PRC. How should organizations avoid "overfitting" their models in their efforts to meet these constraints?

- In Section 3.2, you've left a comment that needs to be removed.

---

> ### Author Response · Authors · 2023-11-15
> **Author Rebuttal**
>
> Thank you for your detailed and very positive feedback on our submission. Below we provide our responses
> to your individual concerns.
>
> Q: Your paper successfully integrates risk measures into the prompt selection process for LLMs.  Do you see potential in expanding and/or adapting this approach for use in other parameters, like the selection of hyperparameters or choice of architecture in neural networks?
>
> A: Yes, while we focus here on prompt selection because we see that as a point of high leverage in the LLM deployment pipeline, these bounding techniques can be used to do model selection with respect to a wide range of hyperparameters, parameters, or even architectures. There are a variety of applications illustrated in the papers from which the bounds are adopted (Angelopoulos et al., 2021, Snell et al., 2022, and Deng et al., 2023) and there has also been some work on applying such techniques to different aspects of large language models (Kumar et al., 2023; Quach et al., 2023; Schuster et al., 2022), although not with respect to prompts or ICL.
>
> Q: How well can the PRC framework adapt to the iterative development and refinement process that characterizes the evolution of large language models?
>
> A: PRC only requires that the loss values are computed once the LLM is fixed, and accommodates any development and refinement process applied to the LLM prior to that. Hence it applies to a particular checkpoint or instantiation of an LLM. Extending the bounding techniques to apply across a set of evolving models across time is an active area of research in DFUQ.
>
> Q: Could the PRC framework inadvertently lead to bias reinforcement? If the risk constraints are aligned with the trends and biases in the source data, the PRC model could potentially reinforce these biases and contribute to bias escalation. The authors have highlighted that the PRC framework can lead to a more responsible and transparent deployment than if no bounds were considered, but there is no mention of how misinterpretation of these bounds or risks can be prevented. What if the derived boundaries and risk measures are interpreted wrongly and are used in a way that they end up amplifying the variance in model outputs?
>
> A: Misinterpreting the bounds is a potential problem (of this and any method providing performance guarantees in deployment). We have outlined the assumptions and limitations of the bounds in the paper, and will ensure they are clear in the final version of the paper.
>
> Q: When risk constraints are not satisfiable, the organization needs to refine the model until it can be controlled by the PRC. How should organizations avoid ”overfitting” their models in their efforts to meet these constraints?
>
> A: The risk of over-fitting in PRC is minimal, as the validation data loss values that PRC operates on must be refreshed whenever the model is updated.

---

> > ### Comment · Reviewer_CoFo · 2023-11-22
> > **Thank you**
> >
> > Appreciate your responses to my questions.

---

### Author Response · Authors · 2023-11-15
**Author Rebuttal**

We thank all reviewers for their useful feedback, and recognizing the importance of including high-probability
upper bounds on diverse risk measures in the pipeline for selecting a prompt and deploying an LLM.

Multiple reviewers stated that the paper was an application of previous techniques to the LLM setting,
with no technical contribution. We would like to highlight our novel proposed algorithm for bounding QBRM
and statistical dispersion measures under distribution shift. The covariate shift setting is of great interest
in both the DFUQ and broader machine learning literature (see responses to reviewers sC6o and FU87 for
specific examples of such research). We believe that our submission offers a significant technical innovation
by enabling the bounding of an extremely rich and important family of risk measures in this setting of great
interest.

---

### Comment · Area_Chair_sh68 · 2023-11-21
**Engaging in discussion with the authors**

Dear reviewers, we are approaching the end of the discussion period (Nov 22) with the authors , please read the rebuttal and engage with authors to discuss any further questions/ clarifications you may have,

Many thanks

AC

---

### Meta-Review · Area_Chair_sh68 · 2023-12-11

**Metareview:**

The paper proposes prompt selection (for example for a zero shot learning task), based on distributional assessment of the incurred loss. The prompt risk control proposed in this paper allows the control of different risk measures such as mean CVaR or Gini Coefficient.  Experiments on chatbots, medical question answering, and summarization illustrate this framework.

The proposed method was appreciated overall by the reviewers but they raised multiple concerns :

Reviewer CoFo while positive about the work mentioned that the paper felt rushed and that the authors don't provide a way to select a risk measures and what are the tradeoffs when considering multiple risks. This question was not addressed by the authors in the rebuttal and we encourage them to discuss it in the paper.

Reviewer YTab raised the question of the expensive cost of the method and its comparison to mean reward that was included in the paper. Reviewer YTab questioned the limitation of the iid assumption to which authors replied that they  address this by extending the quantile-based bounds to the covariate shift setting.

 Reviewer sC6o raised an important question of sample complexity to ensure we have statistical significance. Authors thought it is out of the scope of the work. This question should be discussed and to base it on known results to at least be sure of the significance of the estimates.

The paper would benefit from more polishing and in addressing how to select which risk measure and how to control for multiple risks at hand. The important question of sample complexity should be addressed  to complete the picture.

**Justification For Why Not Higher Score:**

The paper requires further improvements to incorporate reviewers feedback.

**Justification For Why Not Lower Score:**

The use of risk control in LLM deployment is an interesting and under-explored topic.

---

### Decision · Program_Chairs · 2024-01-16

Accept (poster)